# SARS-CoV-2 Is More Efficient than HCoV-NL63 in Infecting a Small Subpopulation of ACE2+ Human Respiratory Epithelial Cells

**DOI:** 10.3390/v15030736

**Published:** 2023-03-13

**Authors:** Gino Castillo, Rahul K. Nelli, Kruttika S. Phadke, Marlene Bravo-Parra, Juan Carlos Mora-Díaz, Bryan H. Bellaire, Luis G. Giménez-Lirola

**Affiliations:** 1Department of Veterinary Diagnostic & Production Animal Medicine, College of Veterinary Medicine, Iowa State University, 1850 Christensen Drive, Ames, IA 50011, USA; 2Department of Veterinary Microbiology and Preventive Medicine, College of Veterinary Medicine, Iowa State University, 1850 Christensen Drive, Ames, IA 50011, USA

**Keywords:** ACE2, *Alphacoronavirus*, *Betacoronavirus*, COVID-19, human respiratory epithelial cells, HCoV-NL63, SARS-CoV-2

## Abstract

Human coronavirus (HCoV)-NL63 is an important contributor to upper and lower respiratory tract infections, mainly in children, while severe acute respiratory syndrome coronavirus 2 (SARS-CoV-2), the etiological agent of COVID-19, can cause lower respiratory tract infections, and more severe, respiratory and systemic disease, which leads to fatal consequences in many cases. Using microscopy, immunohistochemistry (IHC), virus-binding assay, reverse transcriptase qPCR (RT-qPCR) assay, and flow cytometry, we compared the characteristics of the susceptibility, replication dynamics, and morphogenesis of HCoV-NL63 and SARS-CoV-2 in monolayer cultures of primary human respiratory epithelial cells (HRECs). Less than 10% HRECs expressed ACE2, and SARS-CoV-2 seemed much more efficient than HCoV-NL63 at infecting the very small proportion of HRECs expressing the ACE2 receptors. Furthermore, SARS-CoV-2 replicated more efficiently than HCoV-NL63 in HREC, which correlates with the cumulative evidence of the differences in their transmissibility.

## 1. Introduction

Coronaviruses (order *Nidovirales*, suborder *Cornidovirineae*, family *Coronaviridae*, subfamily *Orthocoronavirinae*) constitute a recurring and continuous threat to public and animal health. Human coronavirus infections, like the one caused by a human coronavirus (HCoV)-NL63 (genus *Alphacoronavirus*), are usually mild and associated with only common cold symptoms [1]. However, the consecutive emergence of three members of the genus *Betacoronavirus*, including severe acute respiratory syndrome virus (SARS-CoV) [2], Middle East respiratory syndrome virus (MERS-CoV) [3], and most recently, SARS-CoV-2 (COVID-19 disease) [4] have repeatedly demonstrated the potential of coronaviruses to emerge from animal reservoirs and cause fatal disease in humans. It therefore calls for the development of robust experimental models that could help design mitigations and intervention strategies.

SARS-CoV [5], HCoV-NL63 [6], and SARS-CoV-2 [7] use angiotensin-converting enzyme 2 (ACE2) as the primary receptor for the binding of the viral transmembrane spike (S) glycoprotein for cell entry, an essential step in establishing viral infection. In general, the S protein is cleaved by proteases into S1 and S2 subunits during the maturation process in the infected cells. S1 binds to the ACE2 receptor, whereas S2 mediates viral membrane fusion [8,9]. Meanwhile, ACE2 is a multi-functional molecule expressed across all major tissues of different animal species, and the modulation of its expression is critical for several physiological and pathological processes [10]. The presence and expression levels of ACE2 and how efficiently the virus uses this receptor could determine susceptibility to infection. The similarities between the receptor binding domain (RBD) of HCoV-NL63 and SARS-CoV have revealed a cross-inhibition of the ACE2 transduction by the S protein [11,12]. Thus, HCoV-NL63 has been proposed as a safe BSL-2 surrogate for developing therapeutic interventions against SARS-CoVs.

Often, initial antiviral screening relies on the detection and quantification of replicating viruses in cell culture. Specifically, this study compared susceptibility, replication dynamics, and morphogenesis of SARS-CoV-2 and HCoV-NL63 growing in monolayer cultures of primary human respiratory epithelial cells.

## 2. Materials and Methods

All infection experiments involving SARS-CoV-2 were performed in the BSL-3 laboratory facilities at Iowa State University (ISU) under pre-established/approved protocols.

### 2.1. Primary Human Respiratory Epithelial Cells (HREC)

Commercially acquired primary human respiratory epithelial cells (HREC) (PCS-300-010, ATCC, Manassas, VA, USA) were subcultured on cell culture flasks (Nunc, Thermo Fisher Scientific, Waltham, MA, USA) or plates (Greiner Bio-One North America Inc, Monroe, NC, USA), pre-coated with PureCol^®^ Type I collagen (40 µg/mL/4 mm^2^; Advanced BioMatrix, Inc., San Diego, CA, USA), at a density of ~20,000 cells/cm^2^. HRECs were propagated in ATCC airway epithelial cell basal medium (ATCC) supplemented with 500 mg/mL HSA, 0.6 mM linoleic acid, 0.6 mg/mL lecithin, 6 mM L-Glutamine, 0.4% Extract P, 1.0 mM epinephrine, 5 mg/mL transferrin, 10 nM 3,3’,5-Triiodo-L-thyronine (T3), 5 mg/mL hydrocortisone, 5 ng/mL epidermal growth factor (EGF), 5 mg/mL insulin, 100 IU/mL of penicillin, 100 µg/mL of streptomycin (Pen-Strep) (Gibco™, Thermo Fisher Scientific, Waltham, MA, USA), and 1.25 µg/mL of amphotericin B (AmpB) (Gibco™, Thermo Fisher Scientific) (growth medium #1). Cells were dissociated with 0.5× TrypLE^TM^ express enzyme (Gibco™, Thermo Fisher Scientific) and neutralized using 50% heat-inactivated fetal bovine serum (FBS; EquaFetal™, Atlas Biologicals, Fort Collins, CO, USA), mixed in LHC basal medium (Gibco™, Thermo Fisher Scientific). Collected cells were either seeded directly using respective growth medium or frozen in LHC basal medium containing 30% FBS and 10% dimethyl sulfoxide (DMSO) (Millipore-Sigma, Saint Louis, MO, USA). Specifically, primary cells used in this study corresponded to passages 4–9 for HRECs.

### 2.2. LLC-MK2 Cells

Rhesus monkey (Macaca mulatta) kidney epithelial cells (LLC-MK2 cells; CCL-7™, ATCC) were cultured using growth medium #2 [Eagle’s Minimum Essential Medium (EMEM; ATCC) supplemented with heat-inactivated 10% FBS (ATCC), Pen-Strep, and 25 μg/mL gentamicin (Gibco™, Thermo Fisher Scientific) and incubated at 37 °C with 5% CO_2_.

### 2.3. HCoV-NL63 Culture and Propagation

HCoV-NL63 was propagated in LLC-MK2 cells as previously described [13]. The reagent was deposited by the Centers for Disease Control and Prevention and obtained through BEI Resources, NIAID, NIH: Human Coronavirus, NL63, NR-470. In brief, cells with >90% confluency were pre-incubated at 34 °C with 5% CO_2_ for 1 h prior to virus inoculation. After washing cells twice with phosphate-buffered saline pH 7.4 (PBS) (Gibco™, Thermo Fisher Scientific), HCoV-NL63 was added and incubated for 1.5 h at 34 °C with 5% CO_2_. Subsequently, the inoculum was replaced with growth medium #2 without FBS (infection medium #1) and incubated for 5 days at 34 °C with 5% CO_2_. The virus was harvested by three repeated freeze/thaw cycles at −80 °C, and cell debris was removed by centrifugation at 3000× *g* for 20 min at 4 °C. Approximately 10^5^ 50% tissue culture infectious dose per mL (TCID_50_/mL) of virus titer was achieved, which was aliquoted and frozen at −80 °C for subsequent virus infectious studies on HRECs.

### 2.4. SARS-CoV-2 Culture and Propagation In Vitro

Vero-E6 cells (CRL-1586, ATCC) were used to propagate SARS-CoV-2 according to CDC protocol [14]. The reagent was deposited by the Centers for Disease Control and Prevention and obtained through BEI Resources, NIAID, NIH: SARS-Related Coronavirus 2, Isolate USA-WA1/2020, NR-52281. In brief, cells were subcultured in Dulbecco’s Modified Eagle Medium (DMEM; Corning, Corning, NY, USA) supplemented with Pen-Strep and 10% FBS in a T175 tissue culture flask. At 80% confluency, cells were washed with PBS, and the SARS-CoV-2 virus was added to the cells in a 5 mL volume for 30 min and were shaken every 5–7 min. After 30 min, 30 mL of warm DMEM supplemented with 5% FBS was added to the cells. The inoculated cultures were then incubated at 37 °C in a humidified 5% CO_2_ incubator and observed for daily viral replication and cytopathic effects (CPE). Viral supernatants were collected from cell culture flasks showing CPE greater than 90%, and after removing the cell debris by centrifugation, the virus titer in the supernatants was determined by plaque assay in Vero-E6 cells. After 3 passages of virus propagation, approximately 10^7^ PFU/mL of virus titer was achieved, which was aliquoted and frozen at −80 °C for subsequent virus infectious studies on HRECs.

### 2.5. HCoV-NL63 and SARS-CoV-2 Infectious Studies in HRECs

Viral kinetics in HRECs were performed by seeding 1 × 10^5^ cells per well in collagen-coated 24-well plates (Greiner Bio-one) using growth medium #2; for staining 96-well clear flat bottom black polystyrene surface-treated microplate (CellBIND Costar; Corning) was used, as described previously [15]. After 24 h of incubation, cells were washed twice with 500 μL of LHC base medium (Gibco, Thermo Fisher Scientific), and inoculated at MOI-2 and MOI-1 in triplicate with HCoV-NL63 or SARS-CoV-2 or mock inoculated with infection medium #2 (airway epithelial cell basal medium [PCS-300-030; ATCC] supplemented with 2% Ultroser G [Sartorius, Göttingen Germany], 1% MEM nonessential amino acids solution [Gibco, Thermo Fisher Scientific], 1% HEPES [Gibco, Thermo Fisher Scientific], 1% GlutaMax [Gibco, Thermo Fisher Scientific], 100 IU/mL penicillin, and 100 μg/mL streptomycin) and incubated at 37 °C with 5% CO_2_ for 2 h. Cells were rinsed with LHC basal medium, fresh infection medium #2 was added to each well, and plates were incubated at 37 °C with 5% CO_2_ for 96 h.

### 2.6. Immunohistochemistry Staining in Tissues and Cell Cultures

Immunohistochemistry (IHC) staining was used to confirm the expression of ACE2 in paraffin-embedded tracheal tissue sections from healthy adult donors (commercially acquired via Novus Biologicals, LLC, Centennial, CO, USA) and primary HRECs. Deparaffinized sections were heat retrieved (96 °C for 30 min) using a citrate buffer (Millipore-Sigma) and washed in tris-buffered saline containing 0.1% Tween 20 (TBST) (Millipore-Sigma). For HRECs, confluent cell monolayers on 96-well plates were fixed with 4% paraformaldehyde for 15 min and subsequently permeabilized with 0.1% Triton X-100 (Millipore-Sigma) for 10 min. The following primary antibodies were used in this study: mouse monoclonal anti-ACE2 (4 μg/mL; E-11 Santa Cruz Biotechnology, Dallas, TX, USA); mouse anti-pan-cytokeratin (0.5 μg/mL; AE1/AE3; Bio-Rad Laboratories, Hercules, CA, USA); anti-HCoV-NL63 nucleocapsid (N) protein monoclonal antibody (2D4; Ingenasa-Eurofins, Madrid, Spain) (0.25 µg/mL) and a recombinant anti-SARS-CoV-2 nucleocapsid (N) protein rabbit monoclonal antibody (0.75 μg/mL) (BEI Resources) [The following reagent was obtained through BEI Resources, NIAID, NIH: Monoclonal Anti-SARS Coronavirus/SARS-Related Coronavirus 2 Nucleocapsid Protein (produced in vitro), NR-53791; SinoBio Cat: 40143-R001].

Both tissue sections and cells were stained using an ImmPRESS VR anti-mouse/anti-rabbit IgG HRP polymer detection kit (Vector Laboratories, Newark, CA, USA) following the manufacturer’s instructions. In brief, sections/cells were blocked with animal-free RTU buffer (Vector Laboratories) for 30 min and incubated overnight with the corresponding mouse or primary rabbit antibody at 4 °C. The tissues were treated with 0.1% hydrogen peroxide for 1 h, while cells were treated for 5 min, followed by incubation with the respective secondary antibody for 60 min. Chromogenic detection in situ was performed using ImmPACT DAB EqV peroxidase substrate solution (Vector Laboratories) and hematoxylin, followed by mounting (tissue sections only) in Tissue-Tek Glas mounting medium (Sakura Finetek, Torrance, CA, USA). Microscopic images were captured using an Olympus^®^ CKX4 microscope (Olympus^®^ Corp., Center Valley, PA, USA), Infinity 2 camera, and Infinity Analyze imaging software (Ver 6.5.5, Lumenera Corp., Ottawa, ON, Canada).

### 2.7. Cellular Characterization Using Flow Cytometry

Confluent monolayers of HRECs were trypsinized as described herein, and the dissociated cell suspension was incubated in PBS containing 100 μg/mL bovine deoxyribonuclease I (Millipore-Sigma) and 5 mM magnesium chloride (Millipore Sigma) for 15 min at room temperature. After incubation, the cell suspension was passed through a 30 μm cell strainer (Miltenyi Biotec, Bergisch Gladbach, Germany), and cells were washed thoroughly by centrifugation at 200× *g* for 5 min. Flow cytometric staining was performed using a cell concentration of approximately 2 × 10^5^ cells per treatment in FACS buffer (PBS supplemented with 1% FBS and 0.09% sodium azide). After a 30 min incubation step on the ice and washing twice with FACs buffer, the cells were stained with LIVE/DEAD™ Fixable Near-IR Dead Cell Stain Kit (Invitrogen, Thermo Fisher Scientific, Waltham, MA, USA) at a previously determined concentration of 1:200. In case of ACE2 receptor expression, cells were stained with mouse anti-ACE2 (Santa Cruz Biotechnology) and fixed with BD Cytofix/Cytoperm™ solution (BD Biosciences, San Jose, CA, USA) for 20 min on ice. For assessing the pan-cytokeratin expression, fixed cells were permeabilized with Perm/Wash™ buffer (BD Biosciences) for 30 min on ice, washed, and stained for mouse anti-pan-cytokeratin (Bio-Rad Laboratories). Then, after 30 min incubation on ice with a goat anti-mouse labeled Alexa Fluor^®^ 647 (15 μg/mL, Jackson ImmunoResearch Laboratories, Inc., West Grove, PA, USA), the cells were washed twice and resuspended into 200 μL FACS buffer. Samples were analyzed on an Attune NxT flow cytometer equipped with an autosampler (Thermo Fisher Scientific) according to manufacturer protocols, including the use of appropriate threshold and gate settings. Each sample was tested in duplicate, including unstained, fluorescence minus one (FMO), and isotype controls. Compensation controls were also performed, and the corresponding data were analyzed using the instrument software.

### 2.8. Viral Binding Assay

Human tracheal epithelial section slides were incubated overnight at 37 °C with 250 μL of heat-inactivated SARS-CoV-2 isolate USA-WA1/2020 or HCoV-NL63 at 37 °C in a humidified chamber. After overnight (~16 h incubation with the virus, the tissue sections were vigorously washed with TBST for 15 min at room temperature, and the IHC staining was performed as described in Section 2.5 [15].

### 2.9. Reverse Transcription-qPCR (RT-qPCR) Assay

Viral RNA extractions were performed following the manufacturer’s instructions using the E.Z.N.A.^®^ Viral RNA Kit (Omega Bio-tek, Inc., Norcross, GA, USA). The reverse transcription (RT-qPCR) reactions were performed in triplicate using the following primer/probes listed in Table 1. Each RT-qPCR reaction (20 μL final reaction volume) was set up by combining 3 μL of the extracted template RNA in TaqMan Fast Virus 1-Step Master Mix (Applied Biosystems™, Thermo Fisher Scientific, Foster City, CA, USA), 500/125 nM primer/probes for SARS-CoV-2 or 400/100 nM primer/probes for HCoV-NL63.

All RT-qPCR reactions included the positive control obtained through BEI Resources, NIAID, NIH. The following reagents were deposited by the Centers for Disease Control and Prevention and obtained through BEI Resources, NIAID, NIH: qPCR control RNA from heat-inactivated SARS-CoV-2 (isolate USA-WA1/2020, NR 52347; Genomic RNA from HCoV-NL63, NR-44105, and a “no template” control (NTC). RT-qPCR reactions were run on Rotor-Gene Q (QIAGEN) with cycling conditions of 50 °C for 5 min holding for reverse transcription, 40 cycles of 95 °C for 20 s denaturation, 95 °C for 3 s, and 60 °C for 30 s for amplification. The RT-qPCR results were analyzed using Rotor-Gene Q Pure Detection software (v 2.3.1). For this study, samples with a threshold cycle (Ct) above 35 were considered negative.

### 2.10. Data Analysis

Statistical analyses and plots were performed using the data from flow cytometry, RT-qPCR and analyzed using GraphPad Prism^®^ 9.0.2 software (GraphPad Software Inc., La Jolla, CA, USA) and Excel 365 (Microsoft, Redmond, WA, USA). The statistical significance was determined using the two-way ANOVA multiple comparisons of the Tukey test. For all analyses, a *p*-value < 0.05 was considered statistically significant.

## 3. Results

### 3.1. Distribution of ACE2 Receptor and Virus Binding on the Human Trachea

Human trachea sections stained for ACE2 showed predominant expression on the epithelial cells, particularly towards the tracheal epithelial lining (Figure 1a). However, ACE2 expression was not observed in the subepithelial region of the trachea. Virus binding assay on human tracheal sections incubated overnight with heat-inactivated SARS-CoV-2 (USA-WA1/2020) and HCoV-NL63 revealed virus attachment. Both SARS-CoV-2 and HCoV-NL63 bound on the trachea’s epithelial and subepithelial regions (Figure 1b,c). Meanwhile, the respective secondary antibody controls showed minimal background staining (Figure 1d–f). Despite intrinsic limitations of the IHC technique, background staining was minimized by adjusting different parameters, i.e., antibody concentrations, hydrogen peroxide concentrations, and incubation times with the DAB substrate. In addition, residual specific or non-specific binding of the virus to other cellular receptors in the sub-epithelial region should be considered.

### 3.2. Determining ACE2 Receptors in Human Respiratory Epithelial Cells

Epithelial cell-specific staining of HRECs showed evident expression of pan-cytokeratin (Figure 2a), while the corresponding secondary antibody controls had a minimal background (Figure 2b). Quantification analysis by flow cytometry further confirms that all the cells expressed pan-cytokeratin, but only 6% of these cells expressed ACE2 (Figure 2c).

### 3.3. Demonstration of SARS-CoV-2 and HCoV-NL63 Virus Infection in HRECs by IHC

SARS-CoV-2 virus stocks were generated in Vero-E6 cells, and the SARS-CoV-2 viral nucleoprotein (positive brown staining) was demonstrated in infected cells 72 h post-inoculation (hpi) via IHC staining (Figure 3a). The virus inoculated HRECs with a multiplicity of infection (MOI)-2 (Figure 3b) and -1 were susceptible to SARS-CoV-2 infection, and the virus appeared to replicate efficiently in these cells. Similarly, HCoV-NL63 replicated in LLC-MK2 cells, and corresponding viral nucleoprotein can be observed at 120 hpi (Figure 3c). Again, the HCoV-NL63 virus could infect and replicate efficiently in HRECs at an MOI-2 (Figure 3d) and -1. As expected, the corresponding mock-inoculated controls (Figure 3e–h) remained negative throughout the infection medium inoculation.

Microscopic examination of SARS-CoV-2 infected cells revealed significant morphological changes (CPE), such as rounding of cells, cell detachment, and vacuolation by 72 hpi, which increased over time and in a dose-dependent manner (Figure 3c). In contrast, for HCoV-NL63 infected cells, the CPE was relatively mild with minimal morphological changes regardless of dose and time of infection (Figure 3i).

### 3.4. SARS-CoV-2 and HCoV-NL63 Virus Replication Kinetics in HRECs Using RT-qPCR

To evaluate the differences in viral replication kinetics between SARS-CoV-2 and HCoV-NL63 in HRECs, supernatants from three replicates were collected at different time points (24, 48, 72, 96, 120 hpi) from each inoculated viral dose and evaluated using *N* gene-based RT-qPCR assays. No significant differences in Ct values were observed during infection, either in SARS-CoV-2 or in HCoV-NL63. The Ct values of MOI-1 were higher than MOI-2 in HRECs inoculated with HCoV-NL63 and SARS-CoV-2 (Figure 4a). However, when compared between the viruses, i.e., HCoV-NL63 and SARS-CoV-2, a statistically significant difference (*p* < 0.05) was observed in Ct values at both MOI-2 and -1 (Figure 4b). Ct values of SARS-CoV-2 were comparatively lower than HCoV-NL63 across all the time points tested, resulting in more significant viral replication of SARS-CoV-2 in HRECs.

## 4. Discussion

Viral receptors are major drivers in defining the host range and tissue-specific or cell-type-specific tropism towards viruses. The presence and expression of specific receptors could determine the outcome of the infection. The human trachea sections analyzed during this study expressed ACE2 receptors and spatial distribution on the epithelial cells, particularly towards the tracheal epithelial lining, justifying further in vitro studies. Human respiratory epithelial cells isolated from the tracheobronchial region were used to evaluate the cell susceptibility and virus replication dynamics in HRECs monolayers inoculated with SARS-CoV-2 and HCoV-NL63. Also, it is important to delineate these early cell–viral events to understand the target host cell pathogenesis and differences in transmissibility between different viruses and subsequent variants.

In line with observations from our previous studies, HCoV-NL63 did not cause CPE in monolayers of HREC, while significant CPE was observed in SARS-CoV-2-infected HREC; the progressive loss of cell viability was driven by a cytotoxicity-mediated mechanism [15,17]. SARS-CoV-2 displayed significantly higher efficiency than HCoV-NL63, infecting an extremely low population of HRECs expressing ACE2 receptors. Recent molecular analysis of the ACE2 receptors in different mammalian species, as well as the RBD domain of the spike protein on SARS-CoV-2, HCoV-NL63, and other coronaviruses, have begun to shed light on the prediction of inter-species transmission [18,19].

Similar to what has been reported for SARS-CoV and MERS-CoV but contrary to SARS-CoV-2, HCoV-NL63 replicated more efficiently in standard immortalized cells than in monolayers of primary HRECs [20], which could correlate with the differences on their transmissibility reported in vivo [7]. In fact, HRECs offer some advantages in sustaining SARS-CoV-2 replication compared with standard immortalized cells [21], which are relatively easy to maintain but are not the natural cell target of the virus. Despite the aforementioned differences in cell susceptibility to infection, the assessment of viral growth kinetics showed a slow viral replication, even for SARS-CoV-2 in HRECs monolayers, compared to Vero E6 cells. ACE2 is indeed a critical factor for susceptibility and outcome of the infection but not sufficient for efficient virus replication.

This and previous studies have demonstrated that HCoV-NL63 replication efficiency in vitro is poor, not only compared to low pathogenic coronaviruses [22] or a highly transmissible virus-like SARS-CoV-2 [21], but also when HCoV-NL63 replication is compared across different cells [20] and types of cultures (e.g., monolayer vs. organotypic). Concerning the latter, we demonstrated in a previous study that HRECs and organotypic air-liquid-interface (ALI) HREC, established from the same source (ATCC PCS-300-010; Lot number: 70002486) and the same primary HRECs used in the present study, expressed significantly higher levels of ACE2 receptor protein than HREC cultures and were more susceptible to infection with HCoV-NL63 [17]. It is important to highlight that, contrary to ALI-HRECs, the non-ciliated HREC monolayer cultures lack the morphological, structural, and functional complexity, including ciliated cells, that are required to mimic the respiratory airway in vivo [17]. In this connection, Lee et al. [23] found that the ACE2 receptor was mainly identified within the motile cilia of airway epithelial cells. As reviewed by Bukowy-Bieryllo [24], primary HRECs lack ciliated cells or, after several passages, may lose the ability to form cilia. Moreover, in a recent study [17], our group showed that organotypic air–liquid interface human respiratory epithelial cell (ALI-HREC) cultures in vitro, which resemble the respiratory epithelial lining in vivo, expressed significantly more ACE2 receptor protein than monolayer/non-ciliated HREC cultures. This would explain the apparent discrepancy between ACE2 protein expressions detected in trachea sections (Figure 1) versus HREC (Figure 2). Thus, justifies the development of alternative culture models to assess therapeutic interventions for low pathogenic respiratory coronaviruses like HCoV-NL63, which can be transferable to current and emerging coronaviruses requiring BSL-3 facilities.

## 5. Conclusions

SARS-CoV-2 is more efficient at taking advantage of limited ACE2 receptor availability on monolayer HRECs than HCoV-NL63. Furthermore, SARS-CoV-2 replicated more efficiently than HCoV-NL63 in HRECs.

## Figures and Tables

**Figure 1 viruses-15-00736-f001:**
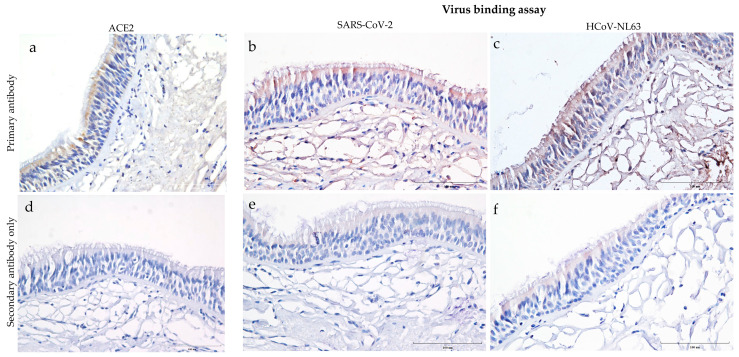
Distribution of ACE2 receptor and attachment of SARS-CoV-2 and human coronavirus (HCoV-NL63) to tissue sections of human trachea. These images show formalin-fixed paraffin-embedded cross-sections of human tracheal sections stained with ImmPRESS VR anti-mouse/rabbit IgG horseradish peroxidase (HRP) polymer detection kit. Dark brown represents the presence of a protein interacting with a specific antibody and is considered a positive expression. Pale brown background, and the nucleus counterstained with hematoxylin is blue. (**a**) Expression of ACE2 in human trachea revealed by immunostaining with mouse anti-ACE2 monoclonal antibody (4 μg/mL); and (**d**) corresponding negative control tissue sections stained with secondary anti-mouse HRP antibody only. (**b**,**c**,**e**,**f**) The tissues sections presented in this panel show virus binding in formalin-fixed paraffin-embedded cross-sections after overnight incubation with 250 μL (**b**) heat-inactivated SARS-Related Coronavirus 2, Isolate USA-WA1/2020 or (**c**) HCoV-NL63. (**e**,**f**) Virus-incubated sections that were stained with secondary anti-rabbit HRP antibody only. Scale bar—100 μm.

**Figure 2 viruses-15-00736-f002:**
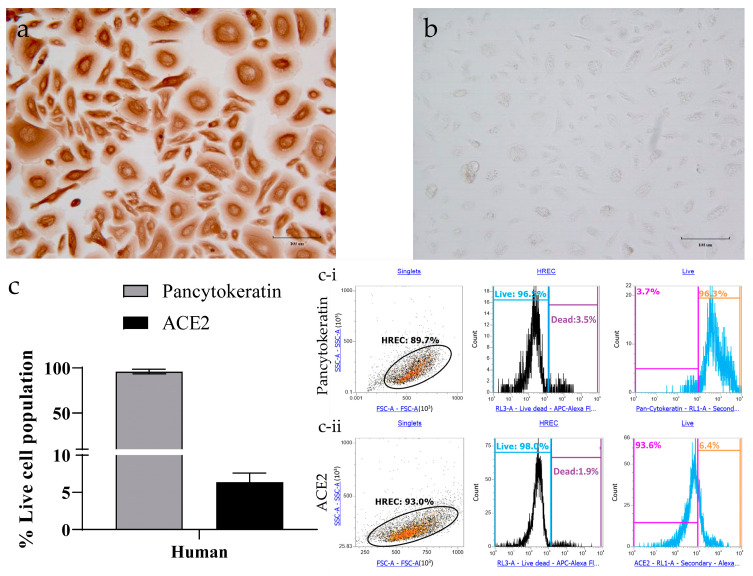
Characterization of human respiratory epithelial cells (HRECs) for pan-cytokeratin and angiotensin-converting enzyme-2 (ACE2) using target-specific markers. Primary HRECs stained for mouse monoclonal to anti-pan-cytokeratin (epithelial cell marker; 0.5 μg/mL) showed strong expression both in (**a**) IHC and (**c**) flow cytometry analysis. (**b**) HRECs incubated with a secondary antibody only displayed minimum background. (**c**) HRECs were also quantified for the ACE2 expression using flow cytometry. Flow cytometry data was collected using an Attune NxT flow cytometer. A representative of 10,000 events were acquired and analyzed for each sample. Cells were gated for singlet population using forward (FSA) and side-scatter (SSA) properties, and the mean of percent live cell population was used to quantify the levels of (**c-i**) pan-cytokeratin and (**c-ii**) ACE2 (*n* = 4). The bar graph represents the mean with the standard error of the mean (SEM).

**Figure 3 viruses-15-00736-f003:**
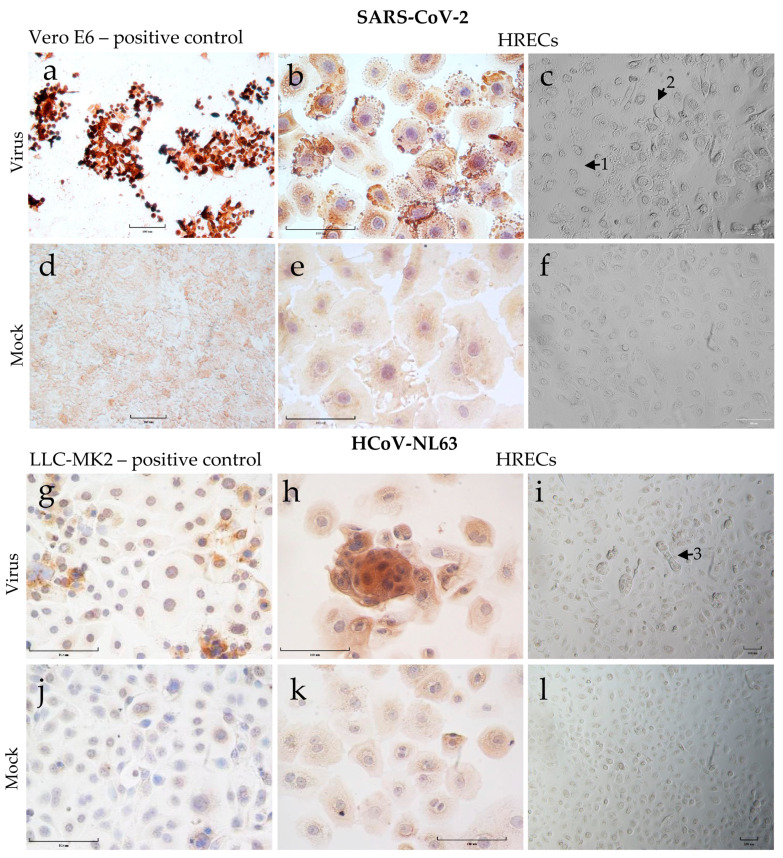
Cytopathic effect (CPE) and detection of viral nucleoprotein (SARS-CoV-2 or HCoV-NL63) in human respiratory epithelial cells (HRECs) and the Vero E6 and LLC-MK2 cell lines used to propagate SARS-CoV-2 and HCoV-NL63. Cells fixed in 4% paraformaldehyde were stained with ImmPRESS VR anti-mouse/rabbit IgG horseradish peroxidase (HRP) polymer detection kit. Anti-SARS Coronavirus/SARS-Related Coronavirus 2 nucleocapsid (N) protein (0.75 μg/mL), and anti-HCoV-NL63 nucleocapsid (N) protein monoclonal antibody (0.25 µg/mL) were used to detect corresponding viruses. Dark brown indicates the presence of a viral protein that interacts with a specific antibody, which is considered a positive expression; pale brown denoted background straining, and the nucleus was counterstained blue with hematoxylin. (**a**) Vero-E6 cells after 72 h post-inoculation (hpi) with SARS-CoV-2 as positive control; (**b**) HRECs after 72 hpi with SARS-CoV-2; (**c**) HRECs showing CPE after 120 hpi with SARS-CoV-2 (1—dying cells, 2—rounding of cells); (**g**) LLC-MK2 cells after 120 hpi with HCoV-NL63; (**h**) HRECs after 120 hpi with HCoV-NL63. (**i**) HRECs showing CPE after 120 hpi with HCoV-NL63 (3—cells mostly healthy looking but with syncytia formation); (**d**–**f**,**j**–**l**) Respective mock inoculations with culture medium. Scale bar—100 μm.

**Figure 4 viruses-15-00736-f004:**
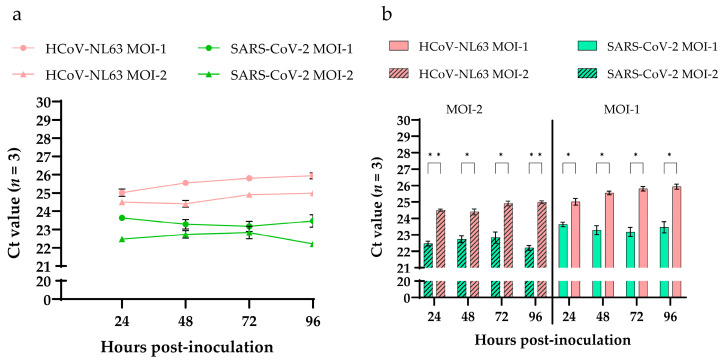
Detection of viral nucleocapsid genes of HCoV-NL63 and SARS-CoV-2 in cell culture supernatants of human respiratory epithelial cells (HRECs). (**a**) Line graph plotted using RT-qPCR Ct values obtained from the supernatants of HRECs inoculated with two doses of virus (MOI-1 and MOI-2) of HCoV-NL63 and SARS-CoV-2; (**b**) Bar graphs showing significantly (*p*-value < 0.05) low Cts in SARS-CoV-2 inoculated HRECs compared to HCoV-NL63 at both MOI-2 and -1 across the time points. All RT-qPCR reactions included mock inoculated, positive, and no-template controls (NTC). Mean Ct values from 3 replicates at each dose were used to plot the graphs, and error bars represent the standard error of the mean (SEM). * denotes *p*-value < 0.05 and ** *p*-value < 0.005.

**Table 1 viruses-15-00736-t001:** List of primer/probe sequences used.

Target Gene	Oligo	Sequence
SARS-CoV-2 N1	Forward	GAC CCC AAA ATC AGC GAA AT
	Reverse	TCT GGT TAC TGC CAG TTG AAT CTG
	Probe	FAM-ACC CCG CAT TAC GTT TGG TGG ACC-BHQ1
SARS-CoV-2 N2	Forward	TTA CAA ACA TTG GCC GCA AA
	Reverse	GCG CGA CAT TCC GAA GAA
	Probe	HEX-ACA ATT TGC CCC CAG CGC TTC AG-BHQ1
HCoV-NL63 N [16]	Forward	GCGTGTTCCTACCAGAGAGGA
	Reverse	GCTGTGGAAAACCTTTGGCA
	Probe	FAM-ATGTTATTCAGTGCTTTGGTCCTCGTGAT-BHQ1
Human RNase P	Forward	AGA TTT GGA CCT GCG AGC G
	Reverse	GAG CGG CTG TCT CCA CAA GT
	Probe	Cy5-TTC TGA CCT GAA GGC TCT GCG CG-BHQ-1

## Data Availability

Not applicable.

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
