# Peer review of "SARS-CoV-2 Is More Efficient than HCoV-NL63 in Infecting a Small Subpopulation of ACE2+ Human Respiratory Epithelial Cells"

_viruses, 2023, doi:10.3390/v15030736_

Round 1
Reviewer 1 Report
Castillo et al., compared the efficiency of SARS-CoV-2 and HCoV-NL63 in infecting human respiratory epithelia cells and found that the former is much more efficient than the latter in infecting the small proportion of respiratory epithelial cells expressing the ACE2 receptor, which is needed for the virus to enter cells. Additionally, SARS-CoV-2 replicates more efficiently than HCoV-NL63 in these cells, which may contribute to its higher transmissibility.
In general, the results are clearly presented and the conclusion is sound. My major concern is that from Fig. 1a it seems the ACE2 is stained in majority of epithelial cells. However, the results from flow cytometry indicate there are only 6% of epithelial cells express ACE2. The authors may want to discuss this inconsistency. In addition, the represented images of flow cytometry should be presented in Fig. 2.
According to the manuscript, epithelial cells infected with SARS-CoV-2 showed severe morphological changes, while those infected with HCoV-NL63 only exhibited mild changes. I believe this is an important observation. However, the authors did not present the result. The should provide image evidence and statistics data (if possible) to support this observation.
Author Response
March 6, 2023
Re: Viruses-2237313 (SARS-CoV-2 is more efficient than HCoV-NL63 in infecting a small subpopulation of ACE2+ human respiratory epithelial cells).
Responses to the reviewer 1
Castillo et al., compared the efficiency of SARS-CoV-2 and HCoV-NL63 in infecting human respiratory epithelia cells and found that the former is much more efficient than the latter in infecting the small proportion of respiratory epithelial cells expressing the ACE2 receptor, which is needed for the virus to enter cells. Additionally, SARS-CoV-2 replicates more efficiently than HCoV-NL63 in these cells, which may contribute to its higher transmissibility.
In general, the results are clearly presented and the conclusion is sound.
Response: Dear reviewer, we thank you for your comments. We feel they have resulted in an improved manuscript and appreciate your efforts on our behalf.
Sincerely,
Dr. Luis G. Giménez-Lirola
My major concern is that from Fig. 1a it seems the ACE2 is stained in majority of epithelial cells. However, the results from flow cytometry indicate there are only 6% of epithelial cells express ACE2. The authors may want to discuss this inconsistency.
Response: The discussion has been modified for further clarification (Line 366-382), including the addition of two new references, yet we would like to nuance a little on your comment. While Figure 1 shows the distribution of the ACE2 receptor and attachment of SARS-CoV-2 and HCoV-NL63 via IHC in formalin-fixed tissue sections of the human trachea, Figure 2 shows flow cytometry results on primary human respiratory epithelial cells (HRECs) obtained from the ATCC. The primary HRECs lack ciliated cells or, after several passages, may lose the ability to form cilia (Bukowy-Bieryllo, Z., (2021). Previously, Lee et al. (2020) found that the ACE2 receptor was mainly identified within the motile cilia of airway epithelial cells. Moreover, in a recent study (Castillo et al. 2022), our group showed that organotypic air-liquid interface human respiratory epithelial cell (ALI-HREC) cultures in vitro, reassembly lining respiratory epithelia in vivo, expressed significantly more ACE2 receptor protein than monolayer/non-ciliated HREC cultures. That could explain the difference in expression levels of ACE2 in tissue sections vs. HRECs (non-ciliated).
Also noticed that the three references above has been included in the discussion.
In addition, the represented images of flow cytometry should be presented in Fig. 2.
Response: Figure 2 has been modified as requested to include flow cytometry images.
According to the manuscript, epithelial cells infected with SARS-CoV-2 showed severe morphological changes, while those infected with HCoV-NL63 only exhibited mild changes. I believe this is an important observation. However, the authors did not present the result. The should provide image evidence and statistics data (if possible) to support this observation.
Response: Figure 3 has been modified to include images showing differential cytopathic effect (CPE) observed in SARS-CoV-2 vs. HCoV-NL63 infected HRECs.
Reviewer 2 Report
This is a pleasant manuscript based on thoroughly designed experiments and solid results, as well as it is carefully and clearly presented. There are some minor mistakes or ambiguities I would like to discuss:
1. In section 2.5, the virus infectious studies in HRECs was performed at 37°C. However, there are some studies showing that temperature can strongly affect the propagation pattern of some CoVs, resulting in changes up to 10-folds level. As the HCoV-NL63 was cultured at 34°C while SARS-CoV-2 at 37°C, indicating that they may prefer different propagating conditions, it is a thorough idea to perform the virus infectious studies at 34°C too. But this is not a requirement and the authors should feel free to take this suggestion or not.
2. In section 3.1, ACE2 expression was not observed in the subepithelial region of the trachea while Both SARS-CoV-2 and HCoV-NL63 bound on the trachea's subepithelial regions. Further explanation or comment should be provided.
3. The results of Figure 2 is sufficient, but I am a little curious why there is no stained picture of ACE2. It will complete Figure 2 by providing that picture, but it is acceptable in the current form.
4. In line 338, the authors talked about inter-species transmission and claim “The inefficient replication of HCoV-NL63 in primary cells could imply a high barrier for inter-species transmission.” This inference is controversial, for there is a counter-argument that this non-ideal interaction between spike and specific specie’s ACE2 may provide a better potential for inter-species transmission of virus because the S protein may bind better to ACE2 of other species.
Author Response
March 6, 2023
Re: Viruses-2237313 (SARS-CoV-2 is more efficient than HCoV-NL63 in infecting a small subpopulation of ACE2+ human respiratory epithelial cells).
Responses to reviewer 2
This is a pleasant manuscript based on thoroughly designed experiments and solid results, as well as it is carefully and clearly presented. There are some minor mistakes or ambiguities I would like to discuss:
Response: Dear reviewer, we thank you for your comments. We feel they have resulted in an improved manuscript and appreciate your efforts on our behalf.
Sincerely,
Dr. Luis G. Giménez-Lirola
- In section 2.5, the virus infectious studies in HRECs was performed at 37°C. However, there are some studies showing that temperature can strongly affect the propagation pattern of some CoVs, resulting in changes up to 10-folds level. As the HCoV-NL63 was cultured at 34°C while SARS-CoV-2 at 37°C, indicating that they may prefer different propagating conditions, it is a thorough idea to perform the virus infectious studies at 34°C too. But this is not a requirement and the authors should feel free to take this suggestion or not.
Response: Correct, according to previous bibliography (Lednicky et al., 2013), HCoV-NL63 is propagated in LLC-MK2 cells at 34°C. However, we performed a pilot experiment comparing both 34°C vs. 37°C. Based on RT-qPCR testing, we found a slight difference between both temperatures (~1 Ct value, lower at 34°C). Then, we made the same experiment with HRECs and, interestingly, we obtained better Ct values using 37°C. Based on this results, and the fact that SARS-CoV-2 is propagated at 37°C, we decided to incubate only at 37°C.
- In section 3.1, ACE2 expression was not observed in the subepithelial region of the trachea while Both SARS-CoV-2 and HCoV-NL63 bound on the trachea's subepithelial regions. Further explanation or comment should be provided.
Response: Although we tried to minimize the background staining by adjusting different parameters such us antibody concentrations, hydrogen peroxide concentrations, and incubation times with the DAB substrate, it is important to highlight some limitations intrinsic to the immunohistochemistry technique. In addition, we cannot rule out some specific or non-specific binding of the virus to other receptors in the cells of the sub-epithelial region. Therefore, section 3.1 has been reviewed for further clarification (Line 242-246).
- The results of Figure 2 is sufficient, but I am a little curious why there is no stained picture of ACE2. It will complete Figure 2 by providing that picture, but it is acceptable in the current form.
Response: Indeed, the expression of ACE2 protein via IHC staining contrast was comparatively lower than pancytokeratin; hence, to confirm our findings, we performed a flow cytometry assay (higher analytical sensitivity), which revealed only 6% of the live cells population positive of ACE2.
- In line 338, the authors talked about inter-species transmission and claim “The inefficient replication of HCoV-NL63 in primary cells could imply a high barrier for inter-species transmission.” This inference is controversial, for there is a counter-argument that this non-ideal interaction between spike and specific specie’s ACE2 may provide a better potential for inter-species transmission of virus because the S protein may bind better to ACE2 of other species.
Response: Agree with your comment. This statement has been removed to avoid confusion (Line 347-349).